# Tokophobia: Psychopathology and Diagnostic Consideration of Ten Cases

**DOI:** 10.3390/healthcare12050519

**Published:** 2024-02-21

**Authors:** Toshinori Kitamura, Mizuki Takegata, Yuriko Usui, Yukiko Ohashi, Satoshi Sohda, Jun Takeda, Tomomi Saito, Yasuyo Kasai, Hideki Watanabe, Megumi Haruna, Satoru Takeda

**Affiliations:** 1Kitamura Institute of Mental Health Tokyo, Tokyo 151-0063, Japan; mtakegata-tky@umin.ac.jp (M.T.); y-ohashi@jiu.ac.jp (Y.O.);; 2Kitamura KOKORO Clinic Mental Health, Tokyo 151-0063, Japan; 3T. and F. Kitamura Foundation for Mental Health Research and Skill Advancement, Tokyo 151-0063, Japan; 4Department of Psychiatry, Graduate School of Medicine, Nagoya University, Nagoya 464-8601, Japan; 5Department of Midwifery and Women’s Health, the Graduate School of Medicine, the University of Tokyo, Tokyo 113-8654, Japan; yusui@g.ecc.u-tokyo.ac.jp (Y.U.);; 6Nursing Faculty, Josai International University, Togane 283-0002, Japan; 7Department of Obstetrics and Gynecology, Graduate School of Medical Science, University of Tsukuba, Tsukuba 305-8577, Japan; 8Department of Obstetrics and Gynaecology, School of Medicine, Juntendo University, Tokyo 113-8421, Japan; jtakeda@juntendo.ac.jp (J.T.); stakeda@juntendo.ac.jp (S.T.); 9Aiiku Research Institute for Maternal, Child Health and Welfare, Imperial Gift Foundation Boshi-Aiiku-Kai, Tokyo 106-0047, Japan; 10Department of Obstetrics and Gynaecology, Japanese Red Cross Medical Center, Tokyo 150-8935, Japan; 11Nagato Clinic, Tokyo 120-0002, Japan

**Keywords:** tokophobia (fear of childbirth), psychopathology, diagnosis, Japanese women

## Abstract

Tokophobia is regarded as the intensive fear of childbirth that some pregnant women have. However, little is known about the psychopathological details of tokophobia (fear of childbirth). Between 2020 and 2021, a total of 10 pregnant women (nine nulliparae and one multipara) with a strong fear of childbirth were referred by obstetricians. Semi-structured psychopathological interviews were conducted, and two cases were judged to have obsession, three an overvalued idea, and one secondary delusion. Three were characterised by both obsession and overvalued idea and one by both obsession and secondary delusion. In total, six cases had features of an overvalued idea. All of the participants except one had a lifetime history of a specific phobia. In addition, their history included social phobia in two cases, panic disorder in one case, obsessive–compulsive disorder (other than tokophobia) in two cases, depressive disorder in two cases, bipolar disorder in two cases, and PTSD in six cases. To conclude, this study showed that tokophobia was not a phobic disorder but a kind of overvalued idea that requires specific assessment and treatment.

## 1. Introduction

Many pregnant women fear their forthcoming childbirth despite their desire to give birth to a baby. In their seminal paper, Hofberg and Brockington (2000) classified tokophobia cases into primary and secondary tokophobia [1]. The former refers to cases of nulliparae while the latter refers to those women who have had a stressful (traumatic) delivery that has triggered tokophobia during their current pregnancy. Thus far, tokophobia, and primary tokophobia in particular, because of its name, has been regarded as a type of specific phobia. There have been, however, few papers discussing the psychopathological aspects of this condition [2]. We start this paper with doubt as to whether tokophobia is actually a type of phobia based on descriptive psychopathology, i.e., phenomenology [3]. We believe that scrutiny of selected cases of severe tokophobia from the perspective of psychopathology will cast a new light on where tokophobia should be placed in the classifications of mental disorders.

A simple discussion may be warranted by comparing a classical phobia and tokophobia. A phobia is usually defined as a fear of a specific object (e.g., animals) or situation (e.g., heights). A person with a phobia almost always feels anxiety when exposed to the external object or situation. Therefore, a person with a phobia is usually active in avoiding the external object or situation. The person is free from anxiety or fear when away from the external object or situation. Women with tokophobia, however, differ in these points. What they fear are birth pain, risks to themselves or the baby, loss of control, and others [4,5]. Nulliparous women have never been exposed to such situations. Their fear is not based on real experiences but on their imagination, which comes from what they have learnt from reading books or hearing stories of women who have given birth. Secondly, pregnant women may well have a chance of avoiding pregnancy, just as a person with a height phobia can work in a ground-floor office, but most pregnant women rather wished to become pregnant. Thirdly, the fear of a phobia is “out of proportion of the actual danger that the object or situation poses, or more intense than it deemed necessary” [6] (p. 199). Is an expectant woman’s fear of labour pain “out of proportion of the actual danger”? These considerations lead us to the next argument: if not a phobia, what psychopathological term should be used to classify the fear of tokophobic women?

Classical psychopathology treats fear as a *form* of thought although its *content* is anxious emotion (as will be discussed later). Therefore, it may be practical to discuss symptomatological characteristics of tokophobia in the psychopathological framework of thought. Looking back at the history of psychopathology for the last century, phobia has not been a stable concept. It has varied from time to time. Jaspers’ (1923/1963) description of phobia is concise: “Patients are beset by an irresistible and terrifying fear of perfectly natural situations and performances: for instance, fear of closed spaces or of crossing open places (agoraphobia)” [7].

The term *phobia* as used in the first half of the 20th century had a wider concept than it has now, encompassing what is now understood as obsession. As early as 1970, Marks (1970) proposed the division of phobic states into Classes I and II. The former is a phobia of stimuli that are *external* to the patient. It includes agoraphobia (wide space), social phobia (meeting or being seen by others), and animal phobia [8]. The latter is a phobia of stimuli that are *internal* to the patient and these cannot be avoided. Class II phobias include illness phobia (e.g., palpitations thought to indicate heart disease, vague pains thought to indicate cancer, that may be termed hypochondriasis or illness anxiety disorder) and obsessive phobia. Interestingly, one of his examples of *obsessional phobia* was the obsessive impulse to kill one’s child. As noted by Marks (1970) [8], Class II phobias were later reclassified as obsessions. Hence, what are feared by individuals with a phobia are things that are located outside of them, such as objects like animals, or situations like heights and seeing blood. Class I fears subside when individuals avoid or are away from the object or have left the place. They no longer feel anxious when the animals are out of their sight, they are on the ground, or they do not see blood. On the other hand, the fears of individuals with Class II phobias are thoughts such as being infected or urges to jump in front of a train (in the absence of the wish to kill themselves). The germs they fear are not in their sight. What they fear is the imaginative idea that they may or will be infected. These feared thoughts or urges, however, are recurrent and persistent despite the individuals’ efforts to ignore them.

What women with tokophobia fear is unique in the framework of phobia and obsession. What is feared is not an external object (like an animal) but a situation (like heights) that is not yet in one’s field of vision. Nulliparous women with tokophobia have not yet endured labour but imagine the situation of labour and labour pain (or still birth). Feared objects and situations in phobic patients are usually avoidable, like animals and high places, whereas childbirth and labour pain are inevitable and therefore unavoidable unless a termination of pregnancy (abortion) is selected. Hence, the fear in tokophobia is more akin to obsession than to phobia (Marks’s Class II) [8]. An analogy may be drawn between the fear in tokophobia and the fear of a jammed door in an elevator in a super high-rise building, or the fear of a roller coaster that has started. The elevator or roller coaster is no longer avoidable. The fear that pregnant women have of future childbirth has been described as the fear of “being at a point of no return” [5] and “the unpredictability of childbirth” [4]. The idea of *the fear of the point of no return* indicates that what is feared is not the *external* object or situation confronting the individual, but what they imagine will appear soon. This explanation may also be applicable to classical phobias. Acrophobic people are afraid because they *expect* to be placed in a high place. Pregnant women with tokophobia are afraid because they *expect* pain and danger due to childbirth.

The pathological content of thinking is classified into (a) obsessive ideas, (b) overvalued ideas, (c) primary and secondary delusions, (d) thought insertion, and (e) others (Table 1). There are several features that distinguish these conditions. These psychopathological categories should be carefully characterised by several features. Obsessions are “thoughts that he [the patient] knows to be his own but which he finds repetitive and strange” [3]. Fear for patients with obsession is their own possession. The domains where obsession takes place include thoughts, images, ruminations, and fears [3]. However, a patient with obsession cannot resist it (difficult to stop thinking about it); there is a lack of perceived control of thought. Obsession “appears against the patient’s will” [9]. Obsession is not ego syntonic; it is not harmonious with the patient’s mood and value systems. The patient, usually, but not always, preserves an insight and is therefore aware of the unreasonable nature of his thoughts. The content of obsessive ideas is strange and beyond reasoning; it can be rectified by others but only for a short time. Other people do not find it difficult to *appreciate* the patient’s fear. Women with tokophobia often know that their fear of labour pain (and/or other things) is beyond what is normally expected and try, but fail, to avoid thinking about it.

Overvalued ideas are “convictions that are strongly toned by affect which is understandable in terms of the personality and its history” [7]. As with those with obsession, patients with overvalued ideas admit that the idea is their own. Unlike obsession, however, the patients feel that they control the idea but do not think the thought is morbid (although unreasonably excessive). They lack insight. They accordingly act on it. Under the rubric of overvalued ideas, McKenna (1984) included, among others, querulous paranoid state, morbid jealousy, hypochondriasis, dysmorphophobia, parasitophobia, and anorexia nervosa. A patient with anorexia nervosa, for example, is characterised by perceptual distortion (“I am fat”), and this conviction is tenacious and resists evidence that is contrary to the conviction (e.g., body weight check). The idea of ‘fatness’ is correctable but only for a short while [10]. There is little insight about morbidity. They do not feel that their idea does not correspond to their mood or value systems. Veale (2002) claimed that overvalued ideas are associated with idealised values, which “have developed into such an over-riding importance that they totally define the self or identity of the individual” [11]. The idealised values of patients with anorexia nervosa are self-control (of weight and body shape) and perfectionism. Some women with tokophobia strongly believe that their fear of the coming labour is ‘justifiable’ and relief from medical assurance does not endure for a long time. Other people, however, may not find it difficult to *appreciate* the patient’s fear.

A delusion is a belief that is not amenable to change in light of conflicting evidence. Patients with delusions are convinced that the idea is their own and correct (no insight). However, the content of a delusion is impossible or false (occasionally bizarre). Other people often, but not necessarily always, find it difficult to *appreciate* the patient’s idea. The distinction between primary and secondary delusions from their origin is important. The former is psychologically irreducible. Jaspers (1923/1963, p. 96) called it “delusion proper”. The latter, “delusion-like ideas”, “emerges understandably from preceding affects, from shattering, mortifying, guilt-provoking or other such experiences, from false-perception or from the experience of derealisation in states of altered consciousness, etc”. (1923/1963, p. 96) [7]. Jaspers repeatedly noted that secondary delusions/delusion-like ideas emerge comprehensibly from other psychic events. They can be traced back psychologically to certain affects, drives, and fears. They are understandable “on the permanent constitution of the personality or of some transient emotional state” [7]. Differentiation between secondary delusions and overvalued ideas is rather vague. It seems a matter of degree rather than a qualitative difference. Jaspers (1923/1963, p. 107) emphasised that overvalued ideas are understandable in terms of the personality and its history. The conviction of fear that some women with tokophobia have is so strong that they do not accept any medical explanation and assurance. Their perception of danger linked to delivery is distorted, e.g., one woman had the absolute conviction that she will die due to an amniotic embolism (“My baby’s birth day is the anniversary of my death” [7].

In thought insertion, the patient “believes that thoughts that are not his own have been inserted into his mind” [12]. Schneider (1959) listed thought insertion as a first-rank symptom that strongly indicates a diagnosis of schizophrenia. In cases of obsessions, overvalued ideas, and delusions, the patients believe that the idea is their own [13]. Thought insertion is different from the above-mentioned categories of psychopathology in that the owner of the thought is not the patient but alien [7,14]. One pregnant woman feared delivery because the idea had been inserted that “the baby she would give birth to was a daughter of the devil” [14]

Little has been studied about the *content* of the mood of tokophobic women. Tokophobia, and secondary tokophobia in particular, is tinted by strong anxiety. Women with a past history of traumatic birth are likely to report fear of the forthcoming childbirth. This may be understood in the framework of the post-traumatic stress theory.

We therefore think that the psychopathological clarification of tokophobia should be preceded by careful phenomenological scrutiny. The present study casts light on the symptomatology of 10 cases. In addition to phenomenological scrutiny, we pay particular attention to the coexistence of psychopathology other than tokophobia. Taking into consideration the complexity of tokophobia symptoms, one empirical means to discuss the concept of tokophobia may be to clarify its association with the present and past history of other mental disorders such as mood, anxiety, and stress-related disorders. If, for example, tokophobic women show higher prevalence of a specified phobia (such as animal phobia), it may be evidence that tokophobia is akin to a specified phobia. On the other hand, the association of tokophobia with past post-traumatic stress disorders may be evidence that tokophobia should be viewed from the perspective of traumatology.

There are many investigations reporting that mood, anxiety symptoms, and disorders are observed among women with tokophobia (e.g., Rondung et al., 2016) [15]. However, most of the past studies reporting the association of tokophobia with anxiety used a general anxiety measure such as the State Trait Anxiety Inventory, which only measures the level of anxiety in general [16,17,18,19,20,21,22,23,24]. Few studies linked tokophobia with specified categories of anxiety disorders. It is of note that although DSM-5 moved obsessive– compulsive disorder (OCD) from the domain of anxiety disorders to that of OCD itself, it has long been recognised that obsession has an emotional element [25].

Clinical studies of tokophobia have long been based either on the use of self-rating questionnaires with multiple items, e.g., [2,16,18,19,21], or on qualitative studies, e.g., [1,5,26]. Little has been studied from the perspective of descriptive psychopathology [3]. This report describes 10 cases of severe tokophobia from the perspective of phenomenology (we did not, although possible, adopt a grounded theory approach here). In addition, the present report pays attention to the past history as well as the current coexistence of diagnosable mental disorders other than tokophobia.

## 2. Materials and Methods

### 2.1. Study Procedures and Participants

This is a secondary analysis of our previous report based on the same 10 cases [26]. The difference is that Takegata and colleagues focused on the narratives of the 10 pregnant women whereas the present report discusses the psychopathological and nosological facets of tokophobia.

A qualitative, descriptive study with individual interviews was undertaken between December 2020 and February 2021.

Pregnant women with a severe fear of childbirth at four obstetric health facilities in Tokyo were asked to participate in this qualitative study. As for the eligibility criteria, pregnant women over the age of 20 with sufficient command of Japanese who were expressing intense fear of childbirth were included. Eligible candidates identified at each clinic were recruited by obstetricians. They were asked to contact two of us (M.T. and T.K.) by email or phone if they were interested. They were informed orally as well as via a written sheet that this would be an interview study about the fear related to childbirth. An informed consent form was sent to each participant’s residence via the postal service prior to the interview day. The informed consent form was submitted to the researcher (M.T. or T.K.) before the interview. Because selection of severe cases of tokophobia was based on clinical judgement of the obstetricians, this was validated by the Japanese Wijima Delivery Expectancy/Experience Questionnaire version A, administered to all the participants before interview.

The interview was conducted by a psychiatrist (T.K.), with the assistance of a midwife (M.T.). T.K. had more 40 years’ experience in clinical psychiatry as well as mental health research with a special interest in perinatal psychiatry. His h index was 48 with 9840 citations (Researchgate 1 February 2024). M.T. had more than 15 years’ experience in midwifery with a special interest in tokophobia. Her h index was 10 with 417 citations (Researchgate 1 February 2024). The interview was conducted as an ad hoc symptomatological structured interview. An English name (alias) was given to each participant temporarily to ensure confidentiality, since the conversation was audio-recorded. Although we initially conducted in-person interviews, asking the participants to come to the Kitamura Institute of Mental Health Tokyo, we changed the interviews to online meetings using Whereby, an online video meeting platform, as the situation of the COVID-19 pandemic became more severe. When online interviews were conducted, we asked the participants to send the signed informed consent form back to the researcher by postal service before the interview date.

### 2.2. Measurements

We conducted diagnostic interviews with the means of an ad hoc structured interview, using the Multidimensional Tokophobia Tool (MTT) [27]. This interview covers a detailed history of the participant’s fear of childbirth and past and present episodes of mood and anxiety disorders. The former includes, in addition to basic obstetric information, (a) a narrative description of the current pregnancy and expected childbirth; and (b) four positive (wonderful, confident, awaiting, and enjoyable) and ten negative (fear of labour pain and medical procedures, lack of confidence, going uncontrolled, lonely, left behind, risky, baby at risk of damage or death, own danger of life, and fear of baby malformation) types of fear of childbirth. We asked the participants their greatest fear out of the list of fears. Then, as a main focus of investigation, we enquired about the psychopathological aspect of the fear. The items included (a) onset of the fear; (b) thought possession (3-point scale); (c) control of thought (3-pint scale); (d) ego dystonicity (3-point scale); (e) insight (3-point scale); (f) correctability (4-point scale); (g) comprehensibility (3-point scale); (h) severity for the last week (percent of the time per day spent feeling fear, range 0% to 100%); (i) disabilities in occupation/house job, interpersonal relationships, hobbies and intrafamilial communication (each rated from 0 to 10); and (j) desire for painless delivery and Caesarean section. The second part covered present and past episodes of mental disorders (other than tokophobia) including (a) specified phobia, (b) social phobia, (c) obsessive–compulsive disorder, (d) panic disorder, (e) agoraphobia, (f) depression, (g) manic/hypomanic episode, and (h) post-traumatic stress disorder (PTSD). Each DSM-5 item of these disorders was asked about in a structured interview format as being either ‘present’ or ‘absent’. Diagnostic syntax followed that of the DSM-5. Onset of all the diagnostic categories was examined if it was during the antenatal or postnatal (within 12 months of the previous childbirth) period. Coding of the MTT was performed mainly by T.K. with discussion with M.T.

### 2.3. Data Analysis

All of the interviews were recorded and their narrative content was analysed in terms of their psychopathological nature as well as the coexistence of other mood and anxiety disorders.

### 2.4. Ethical Consideration

The Institutional Review Board (IRB) of the Kitamura Institute of Mental Health Tokyo approved this study (No. 2019090801). Voluntary participation was ensured by using the informed consent sheet and providing verbal explanations. All electronic and paper-based information collected was carefully stored in a secure location by T.K. to ensure confidentiality.

## 3. Results

### 3.1. Demographics

Our semi-structured interview took approximately 60 min for each participant (Table 2). There were nine nulliparae (one with habitual miscarriage) and one multipara. Their ages ranged between 30 and 42 years old and the mean (SD) of their age was 34.7 (3.9) years. The gestational week ranged between 12 and 38 weeks and the mean (SD) gestation week was 30.2 (8.1) weeks. They were all married and two had partners who were women (one lesbian couple and one bisexual couple).

### 3.2. Target of Fear

The greatest concern was labour pain in six cases and damage to the baby in two cases. Scarlet spent almost all day and Erica spent half the day feeling fearful. The onset of the fear was during the current pregnancy in eight cases, whereas two experienced the onset when they were a university student (Sara) or an elementary school pupil (Alicia).


*“I started to be afraid of childbirth in my elementary school, when I happened to watch horrible scene of childbirth on TV drama”.*
(Alicia)

Despite the fear, some participants did not avoid pregnancy and even received infertility treatments. However, they mentioned that fear of childbirth gradually increased during pregnancy.


*“When I found out I got pregnant, I was delighted because I had received infertility treatment”.*
(Jessica)


*“My fear increased after listening to my friend’s impression that she would not give birth again because the pain was so intense”.*
(Alicia)

### 3.3. Psychopathology of Tokophobia

All the participants recognised that the fear was their own idea (thought possession). None of them felt that the idea was alien. There were, therefore, no cases of thought insertion.

Two cases had difficulty controlling their fear of labour pain and cot death. Another three cases occasionally had difficulty controlling their fear of childbirth. For example, Angelina (age 42 and 35 gestational weeks), who was expecting her first baby after three miscarriages and infertility treatment, was asked, “Fear of labour pain and still birth is your own thought, isn’t it? But, when the thought comes up, can you stop and control it by saying to yourself that it is not the time to think about it?” “I cannot … never ever!” When further questioned, “Is it like an idea, though your own, that comes up spontaneously?” Angelina answered, “I try not to think about childbirth, but I can’t stop doing it. If only I could do it”. When another woman, Scarlet (age 34), expecting her first baby, was asked, “Is it difficult to control your fear? You know, you are not having labour pain right now”. She answered, “Absolutely. There is no pain now”. When further questioned, “There are women who think that it is not an appropriate time to think about it and can stop thinking about it. How about you?” she said, “I could not somehow control it. When there are concerning things, I definitely think about them. This is the case in different matters”. These two women, particularly Scarlet, showed severe difficulty in social functioning. However, the majority of the participants thought that they had control over their fear (control of thought). When asked whether fear would arise when working or talking with friends, Alicia (age 30 and 29 gestational weeks) strongly denied it. Sara (age 31 and 34 gestational weeks) also denied that fear would arise when cooking a meal. Staphany (age 31 and 27 gestational weeks) agreed that she could stop thinking about childbirth when engaging in tasks.

Two women said that fear was not compatible with their life values (ego dystonicity), while eight said that fear was in harmony with their value system (ego syntonicity). An example of ego dystonicity is below.

Interviewer: *“There are women who claim that the idea coming up is different from their own value systems. How about in your case? Is the fear in line with your value systems?”*

Angelina: *“I used to be very optimistic before starting infertility treatment … but when it comes to this pregnancy and childbirth, I am very pessimistic, getting nervous …”.*

Some women showed no insight into the fact that their fear was above that which would be expected by average pregnant women. When asked the same question as above, Sara said, “Never. I never think such fear is nonsense”. Three of the women showed no insight into this.

Correctability was recognised in three cases and one was doubtful. In the other cases, the participants believed that their fear was legitimate. Among these women, the relief of fear due to reassurance by perinatal medical professionals endured only for a very short while.


*“(If other people say that you are too afraid of childbirth, there is nothing to worry about so intensively … do you feel relieved to hear it?) … Hmmm. I would agree with them to start with, and might feel relieved temporarily. … But I would still feel afraid inside of my mind … such advice may be in vain”.*
(Anna)


*“(The same question as above) … I don’t know … I think I would not change myself”.*
(Alicia)

Except for two cases, the content of the women’s fear was appreciable. One of the two cases in which the fear was beyond the interviewer’s appreciation, Rosaline, described her fear as follows.

Interviewer: *“For you, is giving birth to a baby extremely dangerous?”*Rosaline: *“I wrote a will”.*Interviewer: *“Did you?”*Rosaline: *“I have a thought that I will die. However, there have been no laboratory signs of any trouble about me. They say my pregnancy has been smooth. I may be alright. It will be in Japan in well-equipped medical settings where there are professionals …”.*Interviewer: *“Yes, indeed”.*Rosaline: *“Yes, if you see probability, there is virtually none. Nevertheless, completely different is feeling that I will die”.*Interviewer: *“Different, isn’t it? I see the point”.*Rosaline: *“Rationally speaking, it is just one out of ten thousand. I know that. But, my feeling tells me it is one out of two. Probability is high. Although I feel it is high, I persuade myself to calm down taking into consideration that the hospital is good …”.*

Taking into consideration the psychopathological features of different symptom categories related to the fear of childbirth (Table 1), we carefully assessed each case in terms of psychiatric symptomatology. Two cases were judged to have obsession, three cases an overvalued idea, and one secondary delusion. The other cases were more complex. Three were characterised by both obsession and an overvalued idea and one by both obsession and secondary delusion. Hence, six cases had features of an overvalued idea.

Overvalued ideas were the main features of Erica, Sara, and Alicia. Erica’s fear of labour pain started when she became pregnant, progressively becoming severe as the gestational weeks went on. She knew that the fear was her own idea and she could control it (“It is possible to say to myself that it is all right just now”). She said that the fear was not incompatible with her value systems. She had no insight, saying, “People who know my character must not say that my fear is too much”. The relief of her anxiety was short-lived after reassurance by medical professionals. Alicia had a similar fear of labour pain but her fear could not be changed by medical reassurance. Alicia’s intense fear went beyond the level of an overvalued idea and was strongly resistant to medical reassurance. Her fear was not attenuated by medical explanation.

It was difficult to make a differential assessment between obsession and an overvalued idea in three cases (Stephanie, Anna, and Angelina). Stephanie’s fear was ego dystonic with insight, but under conscientious control. Anna’s fear was with insight, but ego syntonic and under conscious control. Angelina’s fear was out of conscious control with insight, but ego syntonic. It was difficult to differentiate between obsession and secondary delusion in Scarlet. Her fear was without control and with insight, but ego syntonic and resistant to medical reassurance.

Therefore, it was hardly possible to identify a single symptomatic term to describe our participants’ fears related to childbirth, and many cases had mixed characteristics of different symptomatological categories. The fear of pain may be a fear of being unable to escape. Alicia said *“It may be my personality … I was afraid of unpredicted phenomena such as earthquake, fire,* etc. *… something that I cannot escape from”).*

### 3.4. Content Emotion Related to Tokophobia

We have thus far discussed the *form* of tokophobia based on classical thought psychopathology. As noted earlier, women with tokophobia feel strong negative emotions while thinking about the delivery to come. Such negative emotion is mainly fear.

The only multiparous woman among our participants, Jessica age 40, expecting her second baby, recalled that her past was characterised by severe labour pain.


*“I was told that induction was necessary because the labour was prolonged for my last childbirth … but pain was extreme … I was afraid of risk to the child”.*


Among nulliparous women, however, some had a past history of traumatic experiences: physical abuse by the mother in childhood (Rosaline), verbal abuse by the mother in childhood (Sara), traffic accident when aged 18 (Anna), and three miscarriages (Angelina). Rosaline, Sara, and Angelina met the full criteria of PTSD. Anna met the threshold of PTSD. Sara recalled the childhood abuse by her mother.


*“The most hurtful was the one when I was three or four years old and my mother was a fulltime housewife and I was so happy with her that I continuously followed her. She shouted, ‘Stop pestering around me!’ That was really threatening. Even when I was a university student, she shouted from outside my room that I did not do house chores. She said, ‘Do not forget annual fees that your parents paid for your university’. That was very saddening”.*


### 3.5. Coexistence of Mood and Anxiety Disorders

All of the participants except one had a lifetime history of a specific phobia. The objects or situations that were the focus of fear varied widely, including darkness (Jessica), diving (Erica), cockroaches (Alicia), butterflies (Rosaline), blood (Sara), dogs (Alicia), insects and injections (Stephanie), anaesthetic injections (Anna), and alligators, snakes, and dirty rivers (Scarlet). All of these phobias started during childhood or adolescence.

The other coexisting psychiatric disorders included social phobia in two cases, panic disorder in one case, obsessive–compulsive disorder (other than tokophobia) in two cases, depressive disorder in two cases, bipolar disorder in two cases, and PTSD in six cases. There were no significant associations between the psychopathological types of fear (obsession, overvalued idea, or secondary delusion) and any category of present or past episodes of psychiatric disorders. An interesting finding was that the participants with secondary delusion, compared to those without it, tended to have a higher rate of past or present episodes of obsessive–compulsive disorder (50% vs. 13%), whereas the participants with an overvalued idea, compared to those without it, tended to have a lower rate of obsessive–compulsive disorder (0% vs. 50%).

## 4. Discussion

The ten participants in our series showed unique clinical pictures. Firstly, they feared the childbirth to come, focusing mainly on labour pain and the possible damages to the baby (including still birth). By definition, the source of the fear is neither an external object nor a situation in which the individual is *currently* located with strong anxiety. The pregnant women were not in the middle of labour but were just *imagining* the pain that would come in due course. Some women had tokophobia some time before the current pregnancy. In 1934, Cates noted “in a primipara, fear of the pains of labour may become an obsession, often intensified by the foolish gossip of well-meaning neighbours, and lead to loss of sleep and digestive disturbances” [28]. Such a feature does not fall into the core of a specific phobia.

All of the participants recognised that the fear they had was their own, not inserted from the outside. There were no cases of hallucinations. Hence, no cases were psychotic. On average, they thought they could have control over the fear (unlike obsession), and that the fear was not incompatible with their life values (unlike obsession), and they had the insight that this was a mental problem (unlike psychosis) but resistant to medical reassurance. The interviewer thought that most of their fears were ‘understandable’ (although difficult to *explain*). Although nine women had a lifetime prevalence of a specified fear (phobia), their childbirth fear was symptomatologically distinct from these specified phobias. Only two cases (Jessica and Rosaline) were characterised as suffering from obsession but four other cases were a mixture of obsession with either an overvalued idea or secondary delusion.

Six cases had features that might be interpreted as overvalued ideas. They were absorbed in their fear, which they felt would match their own value system or personality, they felt they could have control over the fear, and they were reassured by professional explanation (but only for a short time). Hence, there was no sense of ‘intrusion’ of fear. We found it difficult to differentiate secondary delusion from overvalued idea. The only distinction between the two is their degree of conviction that they would be in severe pain or the baby would be hurt. Some women had extraordinarily strong convictions that what they feared would occur.

Many of the participants’ fears were understandable from their life history, mood, and personality. Their life histories were characterised by sexual abuse (Rosaline), harsh discipline (Sara and Stephanie), parental separation followed by foster care (Stephanie), past surgery (Anna), traffic accident (Anna), and repeated miscarriages (Angelina). These findings are in line with previous reports [29,30]. These events may have tinted their way of thinking about themselves, others, and the world. They had past experiences of mood disorder, including two cases of bipolar disorder and PTSD in five cases. Huizink et al. (2004) studied anxiety during pregnancy and found three factors (fear of giving birth, fear of bearing a handicapped child, and concern about one’s appearance) that are distinct from general anxiety and depression [31]. These finding may give support to the notion that fear of childbirth among our series may be a distinctive category and is a type of overvalued idea [11]. Overvalued idea includes many differently named conditions. They are, for example, querulous paranoid state, morbid jealousy, hypochondriasis, dysmorphophobia, parasitophobia, and anorexia nervosa [10]. These share many symptomatological features with tokophobia. We believe that fear of childbirth cannot be fitted to a contemporary classification systems like DM-5 and ICD-11. It should be categorised under the rubric of such terminology. It should be noted that cases with fear are not always given a diagnosis of phobia. We should like to propose the preliminary diagnostic criteria of tokophobia (Figure 1). Here, we propose two specifiers: with obsessive features and posttraumatic features.

Our study no doubt has methodological limitations. Our discussion focused only on the ten cases we interviewed. This may be too small a number with which to reach a conclusion. However, the present study is a mixed approach study, combining quantitative and qualitative methodology. Future studies may try to better express this approach, which is especially useful in topics such as ours. Therefore, our emphasis was on symptomatological (qualitative) analyses. Our argument is one of classical psychopathology. The interview was semi-structured, giving a wide opportunity to the participants to talk freely. More rigidly structured interviews may be needed but we believed that the semi-structured nature might make room for the participants to talk about *deeper* fears and anxieties together with their life history correlates. Another drawback of the present study was the lack of validity and reliability studies for the MTT. Here, training of raters in terms of descriptive psychopathology as well as refinement of the interview and rating format are necessary. Studies with a greater number of participants may give more accurate clinical pictures of tokophobia. In so doing, a questionnaire that covers all of the aspects we have discussed in the present study may be used in a large epidemiological study. The present study was of cross-sectional nature. What remains to be studied is the trajectory of fears and anxieties across the perinatal period, including psychological outcomes after childbirth. For example, fear of childbirth during pregnancy predicts PTSD after childbirth [32,33,34,35,36]. In addition, the distinction of tokophobia from other diagnostic categories requires operational diagnostic criteria. The criteria we presented in Figure 1 are no more than tentative. Here, we defined tokophobia as meeting the clinical pictures described in this manuscript. We added two specifiers: with obsessive features and with post-traumatic features. These two may be added if the case meets the criteria of tokophobia. This proposal should be examined to determine whether what we have proposed is a valid clinical entity.

Clinical implications are multifaceted. In perinatal care, every woman should be assessed in terms of tokophobia and, if present, a therapeutic intervention should be carefully planned and started. As noted in this study, pregnant women with tokophobia are likely to have present or past episodes of mood, anxiety, and stress-related symptoms. These should also be carefully scrutinised so that more appropriate care can be provided [37,38,39,40,41,42]. The presence of tokophobia should not be underestimated because our participants showed difficult social functioning. Practical educational support may be needed in addition to psychotherapy. Women with tokophobia are likely to prefer painless (epidural) delivery or Caesarean section. The request for such a delivery mode may be a hint for perinatal professionals to consider the possibility of tokophobia.

## 5. Conclusions

In conclusion, our study showed that tokophobia was not a phobia but a kind of overvalued idea that requires specific assessment and treatment. Establishing a means to assess facets of tokophobia and diagnostic criteria as well as studies on the natural course of tokophobia are the next steps to take.

## Figures and Tables

**Figure 1 healthcare-12-00519-f001:**
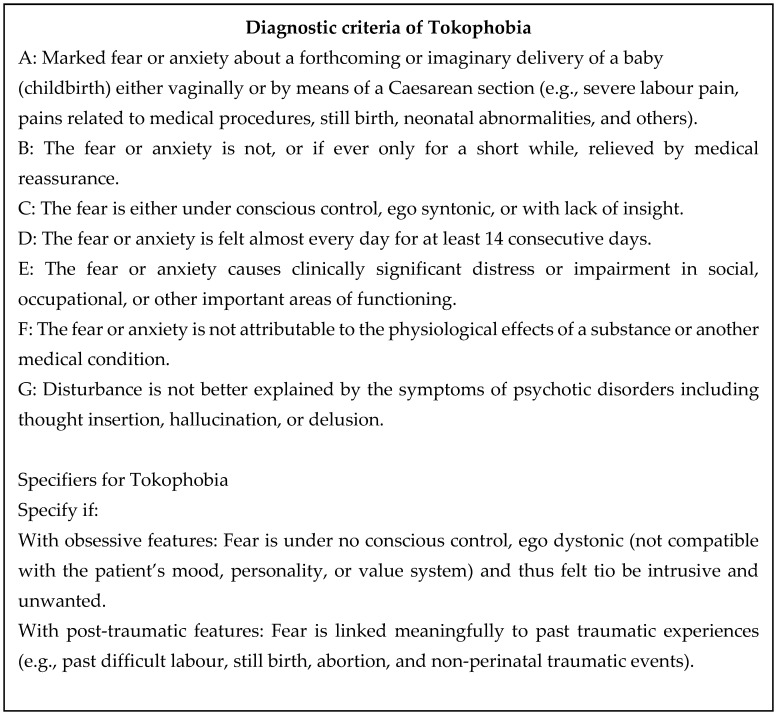
Proposed diagnostic criteria of tokophobia.

**Table 1 healthcare-12-00519-t001:** Classification of thought content psychopathology.

	Obsession	Overvalued Idea	Secondary Delusion	Primary Delusion	Thought Insertion
Thought possession	Own	Own	Own	Own	Alien
Control of thought	No	Yes	Yes	Yes	No
Ego dystonicity	Yes	No	No	No	Yes
Insight	Yes	No	No	No	No
Correctability	Yes	Yes	No	No	?
Comprehensibility	Yes	Yes	Yes	No	No

**Table 2 healthcare-12-00519-t002:** Descriptions of 10 cases.

Case No.	1	2	3	4	5
Pseudonym and age	Jessica, age 40	Erica, age 37	Alicia, age 30	Rosaline, age 34	Sara, age 31
Gestation week	22	33	29	37	34
Greatest concern	Baby will be hurt	Labour pain	Labour pain	Baby will be hurt	Labour pain
Onset	Current pregnancy	Current pregnancy	Current pregnancy	Current pregnancy	University student
Thought possession	Yes	Yes	Yes	Yes	Yes
Control of thought	Yes/No	Yes	Yes	Yes/No	Yes
Ego dystonicity	Yes	No	No	No	No
Insight	Yes	No	No	Yes	No
Correctability	No	Yes/No	No	Yes	Yes
Comprehensibility	Yes	Yes	Yes	No	No
Severity	10	50	20	5	10
Occupation/house job	0	0	0	0	3
Interpersonal relationships and hobbies (range 0 to 10)	8	0	0	0	1
Intrafamilial communication (range 0 to 10)	0	0	0	7	0
Desired delivery model	Epidural	Vaginal	Epidural	Epidural	Caesarean
Parity	Multiparous	Nulliparous	Nulliparous	Nulliparous	Nulliparous
Mood and anxiety disorders (onset age)	Specific phobia (20); OCD (37)	Specific phobia (36); PAN (28); PTSD (23)	Specific phobia (17)	Specific phobia (7); Social phobia (7); Dysthymia (25); PTSD (16)	Specific phobia (12); SOC PHOBIA (13); BP-II (31); PTSD (6)
Symptomatology	Obsession	Overvalued idea	Secondary delusion	Obsession	Overvalued idea
**Case No.**	**6**	**7**	**8**	**9**	**10**
Pseudonym and age	Alicia, age 33	Stephanie, age 31	Anna, age 35	Angelina, age 42	Scarlet, age 34
Gestation week	38	27	35	35	12
Greatest concern	Labour pain	Labour pain	Labour pain	Labour pain/still birth	Labour pain
Onset	Elementary school	Current pregnancy	Current pregnancy	Current pregnancy	Current pregnancy
Thought possession	Yes	Yes	Yes	Yes	Yes
Control of thought	Yes/No	Yes	Yes	No	No
Ego dystonicity	No	Yes	No	No	No
Insight	Yes	Yes	Yes	Yes	Yes
Correctability	Yes	No	No	No	No
Comprehensibility	Yes	Yes	Yes	Yes	Yes
Severity	20	10	5	20	100
Occupation/house job	0	0	0	0	6
Interpersonal relationships and hobbies (range 0 to 10)	0	0	0	8	8
Intrafamilial communication (range 0 to 10)	0	0	0	0	10
Desired delivery model	Epidural	Epidural	Vaginal	Vaginal	Epidural
Parity	Nulliparous	Nulliparous	Nulliparous	Nulliparous	Nulliparous
Mood and anxiety disorders (onset age)	Specific phobia (3)	Specific phobia (3)	Specific phobia injection (13); Subthreshold PTSD	PTSD + MDE (40)	Specific phobia (23); OCD (20); BP-II (23)
Symptomatology	Overvalued idea	Obsession/overvalued idea	Obsession/overvalued idea	Obsession/overvalued idea	Obsession/secondary delusion

BP, bipolar disorder; MD, major depressive disorder; OCD, obsessive–compulsive disorder; PAN, panic disorder; PTSD, post-traumatic stress disorder; SOC PHOBIA, social phobia.

## Data Availability

Due to the nature of this research, the participants of this study did not agree to their data being shared publicly, so supporting data are not available.

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
