# Peer review of "Tokophobia: Psychopathology and Diagnostic Consideration of Ten Cases"

_healthcare, 2024, doi:10.3390/healthcare12050519_

Round 1

Reviewer 1 Report

Comments and Suggestions for Authors

Thank you for the opportunity to review this paper. It is a very interesting topic, and the  authors have put a lot of work in their paper. However, there are some major concerns:

1.       The paper reads as three separate papers: one historical overview, one case series and one clinical proposal.  Although the proposal obviously cannot do without some theoretical introduction and evidence form clinical practice, it seems odd that  a proposal for  a new classifications done, whereas there is limited reference to current classification systems  like DSM5 or ICD-10 in the previous part of the paper. Also references to contemporary literature are lacking. In general references in the introduction are quite old, and sometimes appear quite specific  (e.g.  ref 3) and whereas this is suited for an historical overview, it might be less suitable for introduction a current clinical findings. It may lead to misunderstanding of the theoretical or scientific frameworks the authors choose to start from,  and of the purpose of the paper.

2.       It is important and highly relevant  that the authors make reference to both OCD and PTSD. However that tocophopia may have features of one or both  of these disorders, does not  automatically make that it is not an anxiety disorder anymore. As other authors (eg  Sluijs, A.)  have pointed out, anxiety involves transdiagnostic features as well, in particular avoidance behavior, catastrophizing  and  quality of attention (narrowed, focused only on threat) . Perceived as such, fear of childbirth still seems to match the DSM5criteria of a specific phobia.

3.       It is unclear how lines 92-93, stating that in tocophobia situations are feared, relates to line 104-106, where it is stated that it is is not the situation that  is feared, but  “what they imagine will appear soon”.  But that is still a situation, isn’t it?  I do not understand how this is different from other types of phobias, where fear for what may happen (sooner or later) is also present.  

There are other comments, partly stemming from the points raised above.  However, as the above mentioned points are fundamental to the paper these best be addressed first.

Comments on the Quality of English Language

The texts is quite lengthy, and sentences / paragraphs may be formulated more concise / to the point. 

Author Response

Reviewer #1

  1. The paper reads as three separate papers: one historical overview, one case series and one clinical proposal. Although the proposal obviously cannot do without some theoretical introduction and evidence form clinical practice, it seems odd that a proposal for new classifications done, whereas there is limited reference to current classification systems like DSM5 or ICD-10 in the previous part of the paper. Also references to contemporary literature are lacking. In general references in the introduction are quite old, and sometimes appear quite specific (e.g. ref 3) and whereas this is suited for an historical overview, it might be less suitable for introduction current clinical findings. It may lead to misunderstanding of the theoretical or scientific frameworks the authors choose to start from, and of the purpose of the paper.

Our argument is that cases of tokophobia of which main symptoms are over-valued idea can be fit into none of the contemporary classification systems like DSM-5 and ICD-11. Thus, we changed the expression in Discussion as follows:

These finding may give support for the notion that fear of childbirth among our series may be a distinctive category and is a type of overvalued idea[11]. Overvalued idea includes many differently named conditions. They are, for example, querulous paranoid state, morbid jealousy, hypochondriasis, dysmorphophobia, parasitophobia, and anorexia nervosa[10]. These share many symptomatological features with tokophobia. We believe that fear of childbirth cannot be fit to a contemporary classification systems like DM-5 and ICD-11. It should be categorised under the rubric of such terminology. It should be noted that cases with fear are not always given diagnosis phobia. We should like to propose preliminary diagnostic criteria of tokophobia (Fig. 1). Here, we propose two specifiers: with obsessive features and posttraumatic features.

The point of old references: we think our current knowledge (understanding) is based on the accumulation of the past literature thus far. Psychopathology is particularly the case. Jaspers id old. But this does not imply that his theory is worthless. Careful scrutiny is essential in psychopathological and diagnostic argument.

  1. It is important and highly relevant that the authors make reference to both OCD and PTSD. However that tocophopia may have features of one or both of these disorders, does not automatically make that it is not an anxiety disorder anymore. As other authors (eg Sluijs, A.) have pointed out, anxiety involves transdiagnostic features as well, in particular avoidance behavior, catastrophizing and quality of attention (narrowed, focused only on threat). Perceived as such, fear of childbirth still seems to match the DSM5criteria of a specific phobia.

We agree that a main feature of tokophobia is anxiety. But cases with anxiety symptoms are not always diagnosed as one of anxiety disorders (such as GAD, panic disorder, agoraphobia). OCDs have recently been kicked out of Anxiety Disorders domain. Fear of obesity of an anorexic woman cannot conclude that anorexia nervosa should be categorised as a type of Anxiety Disorder. Our argument is that many cases of ‘tokophobia’ have fears that are psychopathologically classified as overvalued idea (hence fear but not phobia). And there are no specific diagnostic categories in the DSM-5 or ICD-11 that well described the cases of tokophobia. Therefore, we claim necessity of tokophobia as a new diagnostic category.

  1. It is unclear how lines 92-93, stating that in tocophobia situations are feared, relates to line 104-106, where it is stated that it is not the situation that is feared, but “what they imagine will appear soon”. But that is still a situation, isn’t it? I do not understand how this is different from other types of phobias, where fear for what may happen (sooner or later) is also present.

Our argument is that the situation tokophobic women fear is the situation that they have never experienced and that are to come ‘in a moment’ like animal phobia.

Comments on the Quality of English Language

The texts is quite lengthy, and sentences/paragraphs may be formulated more concise/to the point.

We think that a historical review of the literature is essential in this paper.

Reviewer 2 Report

Comments and Suggestions for Authors

Thank you for the opportunity to review this interesting paper which has explored the concept of tokophobia, which is currently defined as a phobia. This has the possibility of altering the diagnosis and support of this disabling condition.

Overall the paper is clear and in most places easy to understand. All sections need to be reviewed for translation errors. Some sentences are missing words or an unusual translation of a word has been used. I have made a couple of comments which I believe will enhance the paper.

Title 

The title is clear.

Abstract 

Provides an overview of the study.

Background

1.       Lines 45-46. I am unsure about what you are saying in this sentence. What is meant by ‘there is doubt as to whether tokophobia would be well suited as a type of phobia…’ Are you saying that tokophobia should not be described as a phobia? The sentence then seems to discussing which methodology is suitable to study this phenomenon. Could you please rewrite or explain what point this sentence is making.

2.       Line 126. What is meant by ‘…strongly toned by affect…’

Methods  

This section was clear and flowed well.  A few more details are required to make the study replicable.

3.       Why were the 4 sites chosen?

4.       There is no mention of any participant information sheet being given to women or what women were informed by the obstetricians or researchers prior to being sent and signing the consent form.

5.       Lines 239 – 242. How were these episodes of mental disorder assessed? Were they questions on participants subjective feelings or screening tools or based on the DSM 5 diagnostic criteria?

Results, discussion and conclusion 

This section would benefit from being more consistent in the way the results are presented.

6.       Line 262. This paragraph went from the previous one talking about demographics into a few sentences about fear. It would flow better if there was a sub heading between the two paragraphs.

7.       Lines 284-305. There are lots of quotes in this section but mixed in with the text whereas the previous and subsequent paragraphs have them as indented quotes which is much easier to read. Have you thought about adding quotes with the name, age and gestation in brackets to avoid so much text and make the paragraph flow?

8.       Line 286. Here you have mentioned the name, age and gestation of a participant but not for all participants. I am not sure of the reason. Could you produce a table with their names, age and gestation or add it at the end of each quote to make the results section consistent.

References 

Many of the references are very old. It is usual to use seminal references but others should be less than 10 years old and preferable newer.

9.       Could some of these be replaced by newer ones. E.g. 30.  Fear of childbirth and history of abuse…, could be replaced with Child sexual abuse and pregnancy: a systematic review of the literature 2020, R BruntonR Dryer.

Table

Table 2. The table is clear but there are numbers for ‘Severity’ and ‘Interpersonal relationships and hobbies’, without knowing the range of scores for these it is difficult to understand the significance. Could you add maximum scores of significant scores to make these sections means something to the reader?

Comments on the Quality of English Language

The majority of the paper has a good command of English. There are some sentences that could be improved. Also there are a few words where an unusual word has been chosen during translation for this context.

Author Response

Reviewer #2

Background

  1. Lines 45-46. I am unsure about what you are saying in this sentence. What is meant by ‘there is doubt as to whether tokophobia would be well suited as a type of phobia…’ Are you saying that tokophobia should not be described as a phobia? The sentence then seems to discussing which methodology is suitable to study this phenomenon. Could you please rewrite or explain what point this sentence is making.

We revised the sentences as follows;

We start this paper with doubt as to whether tokophobia would be well suited as a type of phobia based on descriptive psychopathology, i.e. phenomenology [3]. We believe that scrutiny of selected cases of severe tokophobia from the perspective of psychopathology will cast a new light on where tokophobia should be paced on the classifications of mental disorders.

  1. Line 126. What is meant by ‘…strongly toned by affect…’

This is a quote from Jaspers.

Methods 

This section was clear and flowed well. A few more details are required to make the study replicable.

  1. Why were the 4 sites chosen?

We solicited obstetric clinics that agreed to participate in the study. No other special reason for selection.

  1. There is no mention of any participant information sheet being given to women or what women were informed by the obstetricians or researchers prior to being sent and signing the consent form.

We added the following expressions in Materials and Methods.

They were asked to contact two of us (M.T. and T.K.) by email or phone if they were interested. They were informed orally as well as via a written sheet that this would be an inter-view study about the fear related to childbirth.

  1. Lines 239 – 242. How were these episodes of mental disorder assessed? Were they questions on participants subjective feelings or screening tools or based on the DSM 5 diagnostic criteria?

We added the following sentences in Method.

Each DSM-5 item of these disorders was asked in a structured interview format as either ‘present’ or ‘absent’. Diagnostic syntax followed that of the DSM-5.

Results, discussion and conclusion

This section would benefit from being more consistent in the way the results are presented.

  1. Line 262. This paragraph went from the previous one talking about demographics into a few sentences about fear. It would flow better if there was a sub heading between the two paragraphs.

We inserted two sub headings: 3.1. Demographics of the participants and 3.2 target of fear

  1. Lines 284-305. There are lots of quotes in this section but mixed in with the text whereas the previous and subsequent paragraphs have them as indented quotes which is much easier to read. Have you thought about adding quotes with the name, age and gestation in brackets to avoid so much text and make the paragraph flow?

Yes, we changed expression accordingly adding age and gestational weeks in a bracket.

  1. Line 286. Here you have mentioned the name, age and gestation of a participant but not for all participants. I am not sure of the reason. Could you produce a table with their names, age and gestation or add it at the end of each quote to make the results section consistent.

Yes, we changed expression accordingly adding age and gestational weeks in a bracket.

References

Many of the references are very old. It is usual to use seminal references but others should be less than 10 years old and preferable newer.

  1. Could some of these be replaced by newer ones. E.g. 30. Fear of childbirth and history of abuse…, could be replaced with Child sexual abuse and pregnancy: a systematic review of the literature 2020, R Brunton, R Dryer.

Yes, we replaced the reference [30].

Table

Table 2. The table is clear but there are numbers for ‘Severity’ and ‘Interpersonal relationships and hobbies’, without knowing the range of scores for these it is difficult to understand the significance. Could you add maximum scores of significant scores to make these sections means something to the reader?

Yes, we added the range of the scores.

Comments on the Quality of English Language

The majority of the paper has a good command of English. There are some sentences that could be improved. Also there are a few words where an unusual word has been chosen during translation for this context.

Reviewer 3 Report

Comments and Suggestions for Authors

Dear Authors:

Thank you for allowing me to review this manuscript. It discusses a relevant and important topic that has been the subject of several of my studies. I make some appreciations with the aim of improving this manuscript.

Introduction:

-I find it an interesting introduction, extensive and well documented. I think that it reflects well the need for qualitative studies that provide information that will help to better understand this construct: what can we consider as tocophobia and what can we consider as fear of "normal chidbirth"? This is a frequent discussion among those of us who have studied the fear of childbirth. What is the cut-off point between what would be considered a normal fear and what becomes pathological in a pregnant woman?

I only miss in the introduction to explain that this construct has tried to measure from a quantitative perspective with instruments specifically designed for this purpose (example WDEQ, CFQ etc), not only with generic instruments as you say.

At the end of the introduction should appear more explicitly the objective of this research.  I would like to congratulate the authors for this magnificent introduction.

I understand that this work is part of a research whose results have already been published in a previous paper. Therefore, I am going to make some methodological observations on aspects that may be present in the reference paper, but that should also appear in this one.

-It is a qualitative study, so the methodological approach from which it has been carried out should appear. Has it been carried out from a phenomenological or grounded theory approach? Reading the introduction, both approaches seem possible.

It is not clear how the participants were identified as women with intense fear of chidlbirth. What system or criteria did the obstetricians use for this?

In a qualitative study, Domain 1: Research team and reflexivity (Consolidated criteria for reporting qualitative research (COREQ) is fundamental. This is poorly reflected in this manuscript. Information should be provided on who the research team is, experience in this area, etc.

The semi-structured interview guide should be provided, even if only in supplementary material.

Regarding the Multidimensional Tokophobia Tool, it is an instrument that I do not know and as you reference it, I do not have access to it. Psychometric data on this instrument should be provided, as well as evidence of its validation. This is important because, as I mentioned earlier, there are several instruments used to measure fear of childbirth (both specific and generic). This instrument is quantitative, and why is it used in a qualitative study. If it is quantitative, it should appear in the analysis how the scores have been analyzed. It is not very clear to me how the instrument is used.

This study is a secondary analysis. Okay, but you should explain how it was done (Domain 3: analysis and findings). For example, how many people did the data coding, how were the themes identified, was any statistical software used? There is no data triangulation system. Did participants provide feedback on the findings?

In the last paragraph of the results you speak of non-statistically significant results, which makes no sense in a qualitative study (lines 403-409).

As they state, an important limitation is the number of participants, since category saturation has not been carried out to determine the sample size.

In the same way that I congratulate you for the introduction, I would like to point out that the method should be improved. I believe that this study is really a mixed study, combining qualitative and quantitative methodology. But this would require profound and extensive changes.

Author Response

Reviewer #3

Introduction:

-I find it an interesting introduction, extensive and well documented. I think that it reflects well the need for qualitative studies that provide information that will help to better understand this construct: what can we consider as tocophobia and what can we consider as fear of "normal chidbirth"? This is a frequent discussion among those of us who have studied the fear of childbirth. What is the cut-off point between what would be considered a normal fear and what becomes pathological in a pregnant woman?

I only miss in the introduction to explain that this construct has tried to measure from a quantitative perspective with instruments specifically designed for this purpose (example WDEQ, CFQ etc), not only with generic instruments as you say.

At the end of the introduction should appear more explicitly the objective of this research.  I would like to congratulate the authors for this magnificent introduction.

We added the following sentences at the end of Introduction.

Clinical studies of tokophobia have long been based either on the use of self-rating questionnaires with multiple items [e.g., 2, 16, 18, 19, 21] or on the qualitative studies [e.g., 1, 5, 26]. Little has been studied from the perspective of descriptive psychopathology [3]. This report describes 10 cases of severe tokophobia from the perspective phenomenology (we did not, although possible, adopt a grounded theory approach here). In addition, the present report pays attention to the past history as well as the current coexistence of diagnosable mental disorders other than tokophobia.

I understand that this work is part of a research whose results have already been published in a previous paper. Therefore, I am going to make some methodological observations on aspects that may be present in the reference paper, but that should also appear in this one.

-It is a qualitative study, so the methodological approach from which it has been carried out should appear. Has it been carried out from a phenomenological or grounded theory approach? Reading the introduction, both approaches seem possible.

As described in Study Procedure, we asked obstetricians of four clinics to select pregnant women whom they assessed as severe in fear of childbirth. This was a clinical judgement. We administered the WDEQ to all the participants to validate it. We added the following sentence in Study Procedure.

An informed consent form was sent to each participant’s residence via postal service prior to the interview day. The informed consent form was submitted to the researcher (M.T. or T.K.) before the interview. Because selection of severe cases of tokophobia was based on clinical judgement of the obstetricians, this was validated by the Japanese Wijima Delivery Expectancy/Experience Questionnaire version A administered to all the participants before interview.

It is not clear how the participants were identified as women with intense fear of childbirth. What system or criteria did the obstetricians use for this?

Although a grounded theory approach was possible, we took a phenomenological approach. See above.

In a qualitative study, Domain 1: Research team and reflexivity (Consolidated criteria for reporting qualitative research (COREQ) is fundamental. This is poorly reflected in this manuscript. Information should be provided on who the research team is, experience in this area, etc.

We described the two interviewers in details.

The present interview was conducted by a psychiatrist (T.K.), with the assistance of a midwife (M.T.). T.K. had more 40-year experience in clinical psychiatry as well as mental health research with special interest in perinatal psychiatry. His h index was 48 with 9840 citations (Researchgate 1 February 2024). M. T. had more than 15-year experience in midwifery with special interest in tokophobia. Her h index was 10 with 417 citations (Researchgate 1 February 2024).

The semi-structured interview guide should be provided, even if only in supplementary material.

Regarding the Multidimensional Tokophobia Tool, it is an instrument that I do not know and as you reference it, I do not have access to it. Psychometric data on this instrument should be provided, as well as evidence of its validation. This is important because, as I mentioned earlier, there are several instruments used to measure fear of childbirth (both specific and generic). This instrument is quantitative, and why is it used in a qualitative study. If it is quantitative, it should appear in the analysis how the scores have been analyzed. It is not very clear to me how the instrument is used.

This study is a secondary analysis. Okay, but you should explain how it was done (Domain 3: analysis and findings). For example, how many people did the data coding, how were the themes identified, was any statistical software used? There is no data triangulation system. Did participants provide feedback on the findings?

The Multidimensional Tokophobia Tool (MTT) was an ad hoc structured interview we designed especially for the current research. A main focus of the interview is psychopathological aspects of tokophobia such as thought possession, control of thought, ego dystonicity and others that are assessed in a 3- or 4-point scale. As you requested, we will attach it as a supplementary material. Unfortunately, it is in Japanese. We know it will be unkind for many readers. Its English version is not available right now. Nevertheless, we promise to prepare the English version of the MTT in half a year or so after the present paper is accepted and published. As you expect, coding was performed only by T.K. with discussion with M.T. There are no validation nor reliability studies of the instrument. This is obviously a draw back of this study. It will be important to train raters about descriptive psychopathology specific to tokophobia.

Hence, we rewrote the description of Measurements and added a paragraph in Discussion as a limitation of the study.

We conducted diagnostic interviews by the means of an ad hoc structured interview, the Multidimensional Tokophobia Tool (MTT)[27]. This interview covers a detailed history of the participant’s fear of childbirth and past and present episodes of mood and anxiety disorders. The former includes, in addition to basic obstetric information, (a) a narrative description of the current pregnancy and expected childbirth, (b) four positive (wonderful, confident, awaiting, and enjoyable) and 10 negative (fear of labour pain and medical procedures, lack of confidence, going uncontrolled, lonely, left behind, risky, baby at risk of damage or death, own danger of life, and fear of baby malformation) types of fear of child-birth. We asked the participants their greatest fear out of the list of fears. Then, as a main focus of investigation, we enquired psychopathological aspect of the fear. The items included (a) onset of the fear, (b) thought possession (3-point scale), (c) control of thought (3-pint scale), (d) ego dystonicity (3-point scale), (e) insight (3-point scale), (f) correctability (4-point scale), (g) comprehensibility (3-point scale), (h) severity for the last week (percent of the time per day spent for feeling fear; range 0% to 100%), (i) disabilities in occupation/house job, interpersonal relationships and hobbies, and intrafamilial communication (each rated from 0 to 10), and (j) desire for painless delivery and Caesarean section. The second part covered present and past episodes of mental disorders (other than tokophobia) including (a) specified phobia, (b) social phobia, (c) obsessive-compulsive dis-order, (d) panic disorder, (e) agoraphobia, (f) depression, (g) manic/hypomanic episode, and (h) posttraumatic stress disorder (PTSD). Each DSM-5 item of these disorders was asked in a structured interview format as either ‘present’ or ‘absent’. Diagnostic syntax followed that of the DSM-5. Onset of all the diagnostic categories was examined if it was during the antenatal or postnatal (within 12 months after the previous childbirth) period. Coding of the MTT was performed mainly by T.K. with discussion with M.T.

We inserted sentences (underlined) in Discussion.

Our study no doubt has methodological limitations. Our discussion focused only on the ten cases we interviewed. This may be too small a number in order to reach a conclusion. Therefore, our emphasis was on symptomatological (qualitative) analyses. Our argument is one of classical psychopathology. The interview was semi-structured giving a wide opportunity for the participants to talk freely. More rigidly structured interviews may be needed but we believed that the semi-structured nature might make room for the participants to talk about deeper fears and anxieties together with their life history correlates. Another drawback of the present study was lack of validity and reliability studies of the MTT. Here, training of raters in terms of descriptive psychopathology as well as refinement of the interview and rating format are necessary. Studies with a greater number of participants may give more accurate clinical pictures of tokophobia. In so doing, a questionnaire that covers all of the aspects we discussed in the present study may be used in a large epidemiological study. The present study was of cross-sectional nature. What re-mains to be studied are the trajectory of fears and anxieties across the perinatal period including psychological outcomes after childbirth. For example, fear of childbirth during pregnancy predicts PTSD after childbirth[32-34] [35,36]. In addition, distinction of tokophobia from other diagnostic categories requires operational diagnostic criteria. The criteria we presented in Figure 1 are no more than tentative. Here, we defined tokophobia meeting the clinical pictures descried in this manuscript. We added two specifiers: with obsessive features and with posttraumatic features. These two may be added if the case meets the criteria of tokophobia. This proposal should be examined to determine whether what we have proposed is a valid clinical entity.

In the last paragraph of the results you speak of non-statistically significant results, which makes no sense in a qualitative study (lines 403-409).

We think Reader may wish to know the results of Fisher exact probability. But according to your suggestion, we deleted the part.

As they state, an important limitation is the number of participants, since category saturation has not been carried out to determine the sample size.

We agree. We discussed it in Discussion. Our excuse is difficulty of soliciting participants in the midst of the COVID-19 pandemic.

In the same way that I congratulate you for the introduction, I would like to point out that the method should be improved. I believe that this study is really a mixed study, combining qualitative and quantitative methodology. But this would require profound and extensive changes.

We hope the above-mentioned revision will meet the reviewer’s requirement.

Round 2

Reviewer 3 Report

Comments and Suggestions for Authors

Dear Authors.
Thank you for the work you have done. The manuscript has improved in quality and certain methodological aspects have been notably improved. From my point of view this is a mixed approach study, combining quantitative and qualitative methodology. Future studies can try to better express this approach, which is especially useful in topics such as this one. Let me leave you with some useful references to further explore this topic.
Mixed Methods Research for Nursing and the Health Sciences Editor(s):Sharon Andrew Elizabeth J. Halcomb), GradCert IntCareNurs, PhD, MRCNA, First published:2 January 2009. Print ISBN:9781405167772 |Online ISBN:9781444316490 |DOI:10.1002/9781444316490
Mixed Methods Research: A Guide to the Field (Mixed Methods Research Series Book 3). Vicki L. Plano Clark, Nataliya V. Ivankova. SAGE Publications, 23 Sept 2015 - 368 pp.
For my part can be published the manuscript. Best regards

Author Response

Thank you for your comments. We will read through the references you recommended. Since, however, we have no time before the deadline of submission, I will just add the followings in Discussion.

Our study no doubt has methodological limitations. Our discussion focused only on the ten cases we interviewed. This may be too small a number in order to reach a conclu-sion. However, the present study is a mixed approach study, combining quantitative and qualitative methodology. Future studies may try to better express this approach, which is especially useful in topics such as ours. Therefore, our emphasis was on symptomatolog-ical (qualitative) analyses. Our argument is one of classical psychopathology. The inter-view was semi-structured giving a wide opportunity for the participants to talk freely. More rigidly structured interviews may be needed but we believed that the semi-structured nature might make room for the participants to talk about deeper fears and anxieties to-gether with their life history correlates. Another drawback of the present study was lack of validity and reliability studies of the MTT. Here, training of raters in terms of descriptive psychopathology as well as refinement of the interview and rating format are necessary. Studies with a greater number of participants may give more accurate clinical pictures of tokophobia. In so doing, a questionnaire that covers all of the aspects we discussed in the present study may be used in a large epidemiological study. The present study was of cross-sectional nature. What remains to be studied are the trajectory of fears and anxieties across the perinatal period including psychological outcomes after childbirth. For exam-ple, fear of childbirth during pregnancy predicts PTSD after childbirth[32-34] [35,36]. In addition, distinction of tokophobia from other diagnostic categories requires operational diagnostic criteria. The criteria we presented in Figure 1 are no more than tentative. Here, we defined tokophobia meeting the clinical pictures descried in this manuscript. We added two specifiers: with obsessive features and with posttraumatic features. These two may be added if the case meets the criteria of tokophobia. This proposal should be exam-ined to determine whether what we have proposed is a valid clinical entity.